

# TPX2 upregulates MMP13 to promote the progression of lipopolysaccharide-induced osteoarthritis

Jingtao Yu, Weiqi Wang, Zenghui Jiang and Huashun Liu

Department of Orthopedic Surgery, Zhejiang Hospital, Hangzhou, China

Corresponding author
Huashun Liu, lhsfox@163.com

## ABSTRACT

**Purpose:** This study seeks to identify potential clinical biomarkers for osteoarthritis (OA) using bioinformatics and investigate OA mechanisms through cellular assays.
**Methods:** Differentially Expressed Genes (DEGs) from GSE52042 (four OA samples, four control samples) were screened and analyzed with protein-protein interaction (PPI) analysis. Overlapping genes in GSE52042 and GSE206848 (seven OA samples, and seven control samples) were identified and evaluated using Gene Set Enrichment Analysis (GSEA) and clinical diagnostic value analysis to determine the hub gene. Finally, whether and how the hub gene impacts LPS-induced OA progression was explored by *in vitro* experiments, including Western blotting (WB), co-immunoprecipitation (Co-IP), flow cytometry, *etc.*
**Result:** Bioinformatics analysis of DEGs (142 up-regulated and 171 down-regulated) in GSE52042 identified two overlapping genes (U2AF2, TPX2) that exhibit significant clinical diagnostic value. These genes are up-regulated in OA samples from both GSE52042 and GSE206848 datasets. Notably, TPX2, which AUC = 0.873 was identified as the hub gene. *In vitro* experiments have demonstrated that silencing TPX2 can alleviate damage to chondrocytes induced by lipopolysaccharide (LPS). Furthermore, there is a protein interaction between TPX2 and MMP13 in OA. Excessive MMP13 can attenuate the effects of TPX2 knockdown on LPS-induced changes in OA protein expression, cell growth, and apoptosis.
**Conclusion:** In conclusion, our findings shed light on the molecular mechanisms of OA and suggested TPX2 as a potential therapeutic target. TPX2 could promote the progression of LPS-induced OA by up-regulating the expression of MMP13, which provides some implications for clinical research.

## BACKGROUND

As a degenerative disease, osteoarthritis (OA) is caused by many factors, such as age, obesity, strain, trauma, joint congenital abnormalities, and deformities (*Glyn-Jones et al., 2015*). The disease causes not only labor decline, disability, and premature death, but also a variety of complications with the development of the disease (*Hunter & Bierma-Zeinstra, 2019*). Previous research has indicated that factors such as chondrocyte dedifferentiation, synovitis, and reduced bone density play roles in the pathogenesis of OA (*Loeser, Collins & Diekman, 2016*). However, the precise etiology and pathogenesis of OA remain

incompletely understood. Despite a range of therapeutic options including pharmacological treatments, intra-articular injections, and surgical interventions available to mitigate and slow the progression of OA, a definitive cure for the condition continues to be out of reach (*Sinusas, 2012*). Consequently, identifying therapeutic targets and prognostic markers is crucial to improving clinical outcomes and quality of life in patients with OA disease.

TPX2 microtubule nucleation factor (TPX2) is a target protein of Xenopus plus end-directed kinesin-like protein (Xklp2) containing the TPX structural domain. Although it was once considered a microtubule-interacting protein, it has recently been recognized as an important regulator of proliferation and metastasis in human malignant tumors (*Koike et al., 2022*; *Wang et al., 2022*). The presence of TPX2 was found to trigger accelerated metabolism or clearance of sorafenib, a typical tyrosine kinase inhibitor (TKI), in hepatocellular carcinoma (HCC), leading to resistance of HCC cells to sorafenib (*Wang et al., 2023*). MMPs are produced by a variety of tissues and cells and belong to the family of zinc-dependent protein hydrolases that function to cleave a variety of ECM proteins (*Cui, Hu & Khalil, 2017*). Most MMPs, including matrix metalloproteinase 13 (MMP13), are involved in the turnover of the osteoarticular epithelium, with MMP13 playing a dominant role, especially in processes associated with articular cartilage destruction (*Hu & Ecker, 2021*). It has been demonstrated that iron death occurs in chondrocytes under conditions of inflammation and iron overload. The induction of iron death led to an increase in MMP13 expression in chondrocytes along with a decrease in collagen II expression (*Yao et al., 2021*). Although many researchers have investigated the function of TPX2 and MMP13 in a variety of cancers and diseases, their specific functions in lipopolysaccharide (LPS)-induced osteoarthritis remain largely unexplored.

The gene expression profile array is a data table that depicts the gene expression type and abundant information of a specific cell or tissue in a specific state (*Asp, Bergenstrahle & Lundeberg, 2020*). The identification of gene markers for tumor classification and typing, along with potential targets for pharmacological intervention, is essential in establishing an effective classification model. Advancements in second-generation high-throughput sequencing technologies, including 454, Solexa, SOLID, and HeliScope sequencing (*Pareek, Smoczynski & Tretyn, 2011*; *Choi, 2016*), mark a significant leap from first-generation methods. These technologies are instrumental in elucidating genome structure and function, offering rapidity, high precision, and cost-effectiveness (*Chan, Naphtali & Schellhorn, 2019*). Microarray technology serves as a powerful tool to comprehensively assess gene expression profiles in tissues and provide insight into the multiple factors that control the regulation of gene transcription. The technology not only provides a wealth of information covering markers and predictors that may characterize specific clinical manifestations, but also holds the promise of new applications for therapeutic areas. The scope of this technology extends significantly, offering robust support for advancing disease research and therapeutic development (*Masotti et al., 2010*). Utilizing these cutting-edge methodologies, researchers are empowered to discern differentially expressed genes (DEGs) through gene expression profile analysis, thereby

facilitating the identification of potential biomarkers for various diseases (*Costa-Silva, Domingues & Lopes, 2017*).

In this study, we employed bioinformatics and experimental approaches to investigate biomarkers associated with OA, to offer valuable insights for further advancements in OA research. This study contributed to the study of OA pathogenesis and facilitated the progression of improved diagnostic and treatment targets.

## MATERIALS AND METHODS

### Extraction of microarray data information

The Gene Expression Omnibus (GEO; https://www.ncbi.nlm.nih.gov/geo/), established in 2000, serves as a pivotal repository for high-throughput genomic data. In this study, we focused on the GSE52042 profile for OA from the GEO dataset, encompassing eight samples: four from diseased (as the case group) and four from less affected individuals (as the control group). The profile was stored in TXT format for further analysis. Similarly, another OA-related GSE206848 profile was downloaded, which included 16 samples (seven normal samples, seven OA samples, and two rheumatoid arthritis (RA) samples), among which we selected seven normal samples as well as seven OA samples for analysis.

### Identification of DEGs related to OA

GEO2R is a microarray data analysis tool mainly for gene differential expression analysis based on microarray data (*Barrett et al., 2013*). To select DEGs from eight groups of samples in the GSE52042 profile, we used fold change (FC) >1.5 as the screening criteria for the up-regulation of DEGs, and FC < 0.67 for the down-regulation of DEGs. These thresholds were set to ensure a balanced sensitivity and specificity in detecting gene expression alterations, thereby minimizing false positives and negatives. It indicated statistical significance if $P < 0.05$. Subsequently, the DEGs results were visualized using a Volcano plot in the R programming language.

### Gene ontology and Kyoto Encyclopedia of Genes and Genomes analysis

In this study, we clarified the GO terms and KEGG pathways of DEGs based on the Database for Annotation, Visualization and Integrated Discovery (DAVID; https://david.ncifcrf.gov/) (*Dennis et al., 2003*), and used $P < 0.05$ as the standard for enrichment analysis.

### Construction of protein-protein interaction network and verification of gene expression

The Search Tool for the Retrieval of Interacting Genes (STRING) database (http://string-db.org) is a search tool for studying protein interactions, which can help users quickly search for protein functions. In our research, DEGs data were first imported into the STRING database, the protein-protein interaction (PPI) network was then generated using the Degree algorithm in the Cytoscape software. From this network, the top 15 most

critical genes were selected for heatmap visualization and expression analysis in the GSE52042 and GSE206848 datasets. Subsequently, the "ggvenn" package in R was employed to identify significantly expressed overlapping genes among these 15 for further analysis.

## Gene set enrichment analysis

Gene set enrichment analysis (GSEA) is a computational method employed to analyze and interpret gene expression data from microarray studies. The Molecular Signature Database (MSigDB) is a collection of functional genes specially collected for GSEA analysis and can be directly used for GSEA analysis. In our research, we utilized GSEA, in conjunction with the MSigDB, to conduct Kyoto Encyclopedia of Genes and Genomes (KEGG) pathway enrichment analysis of two overlapping genes identified within the GSE52042 and GSE206848 datasets. Pathways that met the threshold of $P < 0.05$ were deemed statistically significant.

## Receiver operating characteristic analysis on overlapping genes

To assess the clinical diagnostic potential of the overlapping genes in OA, receiver operating characteristic (ROC) curve analysis was performed. Area Under the Curve (AUC) values, along with 95% confidence intervals (CIs), were calculated for the three genes within the GSE52042 and GSE206848 datasets utilizing the "timeROC" package in R software (version R-4.1.2-win). Larger AUC values indicate greater diagnostic accuracy for OA, allowing a measure of the potential utility of gene in a clinical setting.

## Cell culture and treatment

Normal chondrocyte from human cell line C28/I2 (American Type Culture Collection) was cultured at 37 °C in DMEM/F12 medium supplemented with 10% FBS in a humidified environment with 5% $CO_2$. For lipopolysaccharide (LPS) treatment, $5 \times 10^5$ cells were removed to 6-well plates, and various doses of LPS (0, 1, 5, and 10 μg/mL) were applied for 24 h.

## Cell transfection

For TPX2 knockdown, cell transfection was conducted with small interfering RNA (siRNA) targeting TPX2 or negative control (NC) by Lipofectamine 2000 reagent (Invitrogen, Carlsbad, CA, USA). For overexpression of MMP13, cells were transiently transfected with overexpression plasmid encoding MMP13 or empty vector through Lipofectamine 2000 reagent.

## Quantitative real-time polymerase chain reaction assay

We employed TRIzol reagent (Invitrogen, Carlsbad, CA, USA) to collect total RNA in cells. Complementary DNA (cDNA) was synthesized by PrimeScript RT reagent Kit (Takara Bio Inc., Shiga, Japan). qRT-PCR was conducted by SYBR Premix Ex Taq II (Takara Bio Inc., Shiga, Japan) on a LightCycler 480 instrument (Roche, Basel, Switzerland). The primer sequences used were as follows: TPX2 forward: 5′-AGGGGCCCT TTGAACTCTTA-3′, reverse: 5′-TGCTCTAAACAAGCCCCATT-3′; GAPDH forward:

5′-TGACTTCAACAGCGACACCCA-3′, reverse: 5′-CACCCTGTTGCTGTAGCCAAA-3′. The qRT-PCR data were analyzed by the $2^{-\Delta\Delta Ct}$ method.

## Western blotting assay

Protease and phosphatase inhibitors were added to RIPA buffer (Beyotime Biotechnology, Jiangsu, China), which was used to lyse the cells. A BCA protein assay kit from Thermo Fisher Scientific (Waltham, MA, USA) was applied for protein content measurement. SDS-PAGE was used to separate equal quantities of protein, which was then deposited onto PVDF membranes from Millipore. After blocking with 5% non-fat milk, the membranes were incubated overnight at 4 °C with primary antibodies against TPX2 (ab252944; Abcam, Cambridge, UK), MMP13 (ab315267; Abcam, Cambridge, UK), cleaved caspase-3 (ab32042; Abcam, Cambridge, UK), or GAPDH (ab8245; Abcam, Cambridge, UK), all diluted at a 1:1,000 concentration. The protein bands were detected using Thermo Fisher Scientific's ECL substrate after being incubated with an HRP-conjugated secondary antibody, and their quantities were calculated using ImageJ software.

## Cell counting kit-8 assay

The Cell Counting Kit-8 (CCK-8; Dojindo, Mashiki, Japan) was employed to measure cell viability. In 96-well plates, $5 \times 10^3$ cells were put into each well, and LPS treatment was applied for 24 h. Then, 10 μL CCK-8 solution was added to each well and incubated at 37 °C for 2 h. The absorbance was detected by a microplate reader at 450 nm.

## Flow cytometry

Cell apoptosis was assessed using flow cytometry with Annexin V-FITC/PI labeling. Cells ($5 \times 10^5$ cells/well) were seeded in a six-well plate and treated accordingly. After treatment, cells were harvested by centrifugation at 1,500 rpm for 5 min and washed thrice with $1 \times$ PBS. Subsequently, cells were incubated with 5 μL of FITC-conjugated Annexin V and 5 μL of PI for 10 min at room temperature in the dark. The stained cells were swiftly analyzed using a BD FACS Aria II flow cytometer (BD Biosciences, San Jose, CA, USA) for the evaluation of apoptosis.

## Enzyme-linked immunosorbent assay

As directed by the manufacturer, to measure the content of inflammatory factors, IL-6 (# CSB-E04640r; CUSABIO, Wuhan, China), IL-1β (# CSB-E08055r; CUSABIO, Wuhan, China), TNF-α (# CSB-E11987r; CUSABIO, Wuhan, China) enzyme-linked immunosorbent assay (ELISA) kits were utilized. LPS was administered to the cells or an overexpression plasmid was transfected. IL-6, IL-1β, and TNF-α levels in the supernatant were assessed after 24 h.

## Co-immunoprecipitation assay

TPX2 and MMP13 interaction was detected by co-immunoprecipitation using the Pierce Co-Immunoprecipitation Kit (Co-IP; Pierce, Waltham, MA, USA). Cells were processed with LPS and lysed with IP lysis buffer. The cells were lysed using an IP buffer

supplemented with protease inhibitors. Subsequently, the cell lysate underwent pre-cleaning with 20 µl of DynabeadTM Protein A (10001D; ThermoFisher Scientific, Waltham, MA, USA) for 2 h at 4 °C. After pre-cleaning, the cell lysate was further incubated with either the antibody or IgG at 4 °C overnight. Following this incubation, 20 µL of washed magnetic beads were introduced into each reaction and allowed to incubate for an additional 2 h at 4 °C. The co-precipitated complex was then washed with IP buffer and subjected to boiling in the SDS loading buffer. Next, the lysates were kept with primary antibodies against TPX2 or MMP13, followed by protein A/G magnetic beads. The immunoprecipitated complexes were eluted and analyzed by WB.

## Cell nuclear and cytoplasmic extraction

The cytoplasmic and nuclear fractions were separated from LPS-treated human normal chondrocytes using the Thermo Scientific NE-PER Kit (Cat. PI-78833; Waltham, MA, USA). The fractions were then analyzed by Immunofluorescence staining analysis to determine the co-localization of TPX2 and MMP13.

## Immunofluorescence staining

Immunofluorescence staining was conducted to investigate the distribution of TPX2 (ab252944; Abcam, Cambridge, UK) and MMP13 (ab315267; Abcam, Cambridge, UK) in the LPS-treated human normal chondrocytes. The cells were permeabilized with 0.1% Triton X-100, fixed by 4% paraformaldehyde, and blocked with 5% BSA. TPX2 and MMP13-specific primary antibodies were then used to incubate the cells, followed by the proper secondary antibodies. DAPI was used to stain the nuclei. With the use of a fluorescent microscope, pictures were taken.

## Statistical analysis

Data were expressed as mean standard deviation (SD), and each experiment was carried out in triplicate technical replicates. The statistical analysis was done with the aid of GraphPad Prism 8.0. Statistical significance was indicated by a $P$-value $< 0.05$.

# RESULTS

## Screening of GSE52042-DEGs and enriched terms and pathways

To identify DEGs in OA patients, we analyzed samples from the GSE52042 gene expression profile using the GEO2R tool under predefined conditions. This analysis yielded 142 up-regulated and 171 down-regulated DEGs (Fig. 1A). These DEGs were enriched in biological process (BP, Fig. 1B) in GO in keratan sulfate catabolic process, organelle assembly, and so on. For cell components (CC, Fig. 1C), the DEGs were related to the microtubule, nucleus, cytoskeleton, *etc*. For molecular function (MF, Fig. 1D), these DEGs were associated with unmethylated CpG binding, siRNA binding, single-stranded DNA blinding, *etc*. KEGG analysis demonstrated the first 10 pathways that DEGs enrichment were the VEGF signaling pathway, malaria, adherens junction, progesterone-mediated oocyte maturation, cAMP signaling pathway, *etc*. (Fig. 1E).

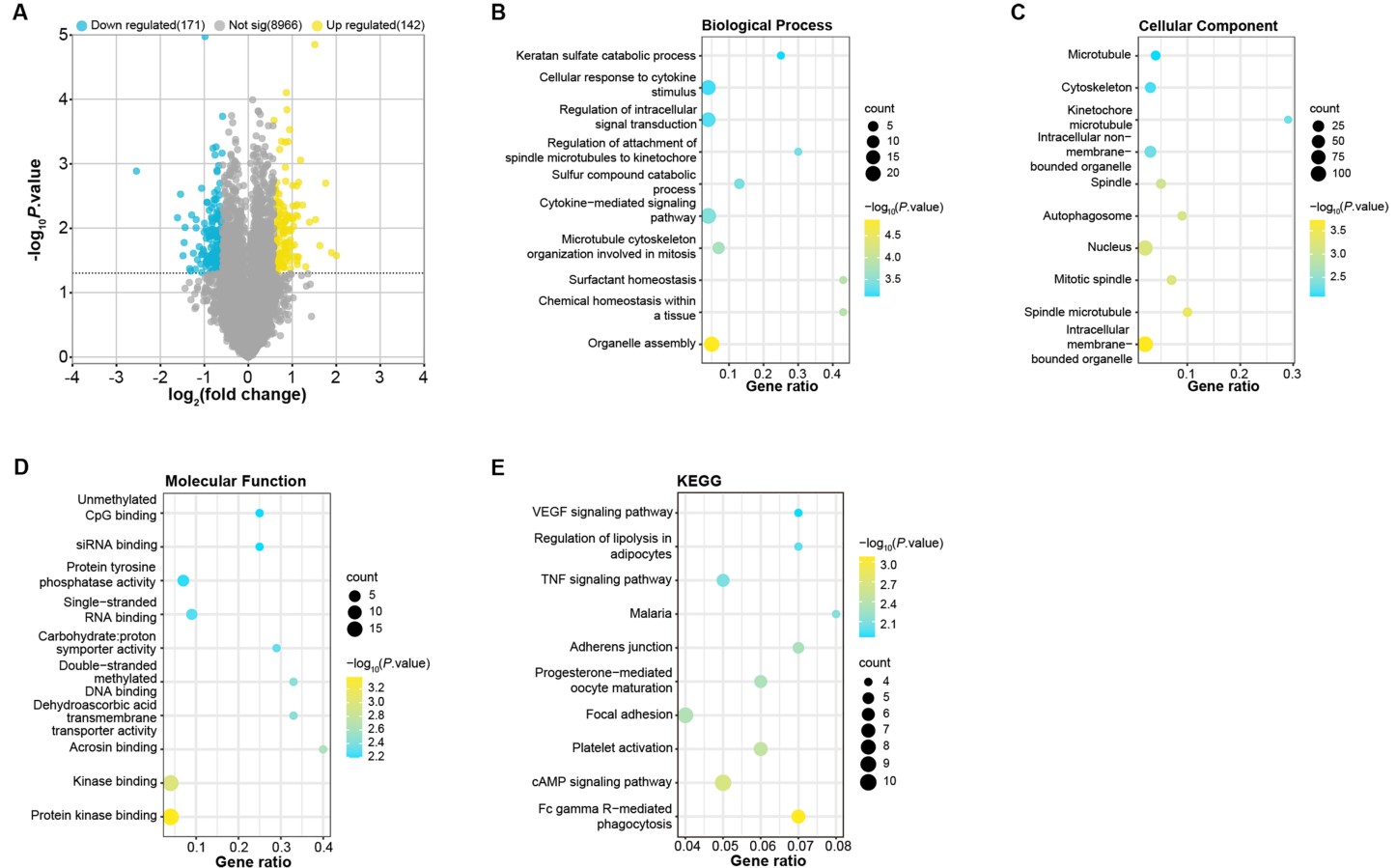

**Figure 1** Identification and enrichment analysis of DEGs in the GSE52042 dataset.

## TPX2 and U2AF2 were identified as overlapping genes

The constructed PPI network comprised 229 nodes and 453 edges (Fig. 2A). Based on gene degree values within this network, the top 15 genes were identified, including AURKA (degree = 22), CCNB1 (degree = 20), TPX2 (degree = 19), RACGAP1 (degree = 19), CDC25C (degree = 18), NEK2 (degree = 17), CENPE (degree = 16), CENPN (degree = 16), TRIP13 (degree = 16), BIRC5 (degree = 15), DHX9 (degree = 15), U2AF2 (degree = 14), KIF18A (degree = 13), SKA3 (degree = 13), CDT1 ( degree = 13) (Fig. 2B). We then performed a heat map analysis on the GSE52042 and GSE206848 to check the batch effects of the two datasets (Figs. 2C and 2D). Subsequent comparative analysis revealed that, in the GSE52042 dataset, these 15 genes were generally significantly overexpressed in OA samples (Fig. 2E). In the GSE206848 sample, only the expressions of NEK2, TPX2, CENPE, DHX9, and U2AF2 were significant (Fig. 2F). According to this result, the "ggvenn" package screened two overlapping genes from these genes with the same significant expression results in the two datasets, namely TPX2 and U2AF2 (Fig. 2G).

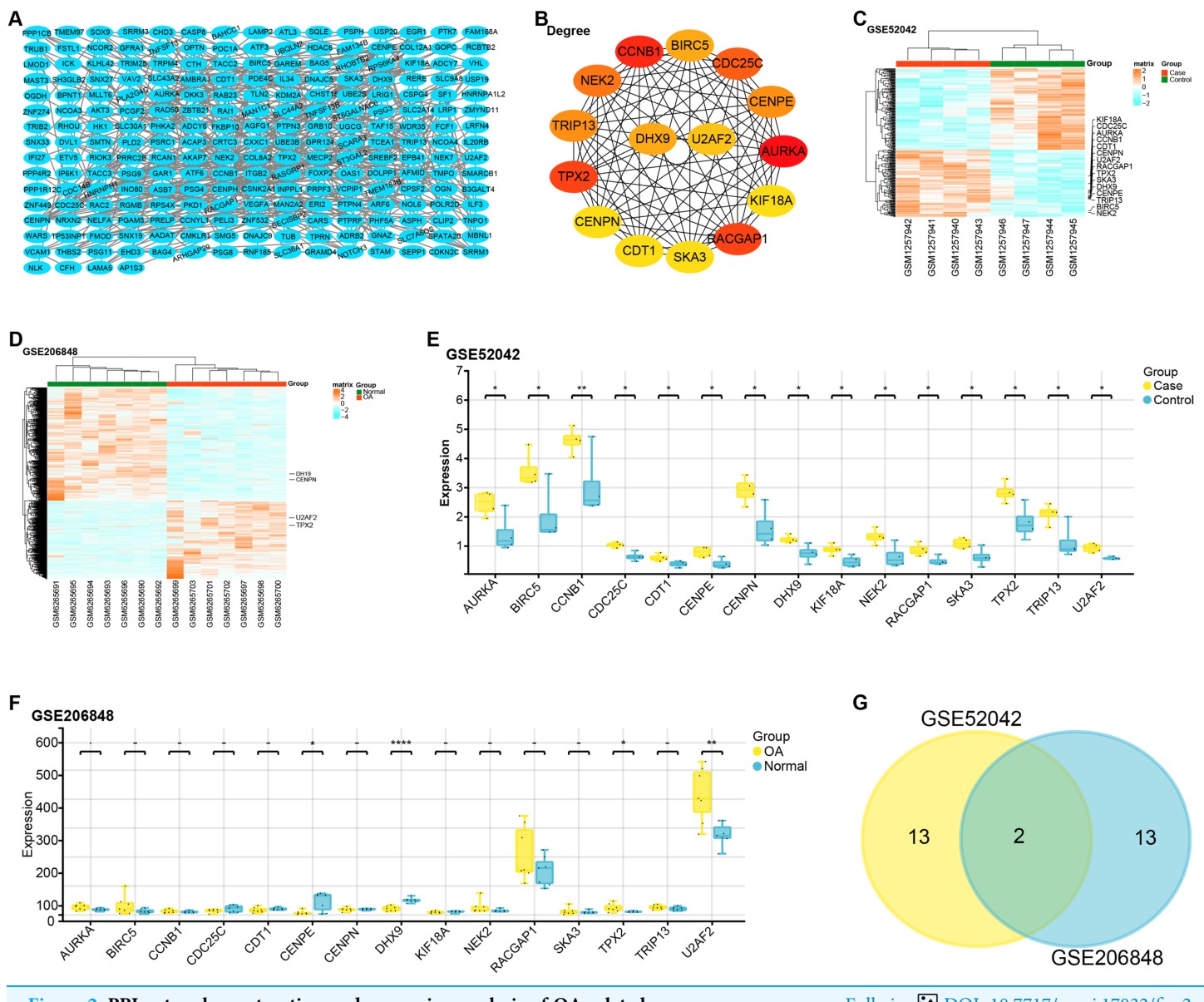

**Figure 2  PPI network construction and expression analysis of OA-related genes.**

## Pathway enrichment analysis of overlapping genes in GSE52042 and GSE206848 datasets

We conducted a GSEA-KEGG pathway enrichment analysis on overlapping genes (Figs. 3A–3D). In the GSE52042 dataset, TPX2 was associated with starch and sucrose metabolism, mismatch repair, and lysosome, while U2AF2 was enriched in the p53 signaling pathway, apoptosis, *etc*. In the GSE206848 dataset, TPX2-enriched KEGG pathway enrichments included basal transcription factors, WNT signaling pathway, and U2AF2-enriched pathways include prion diseases, *etc*.

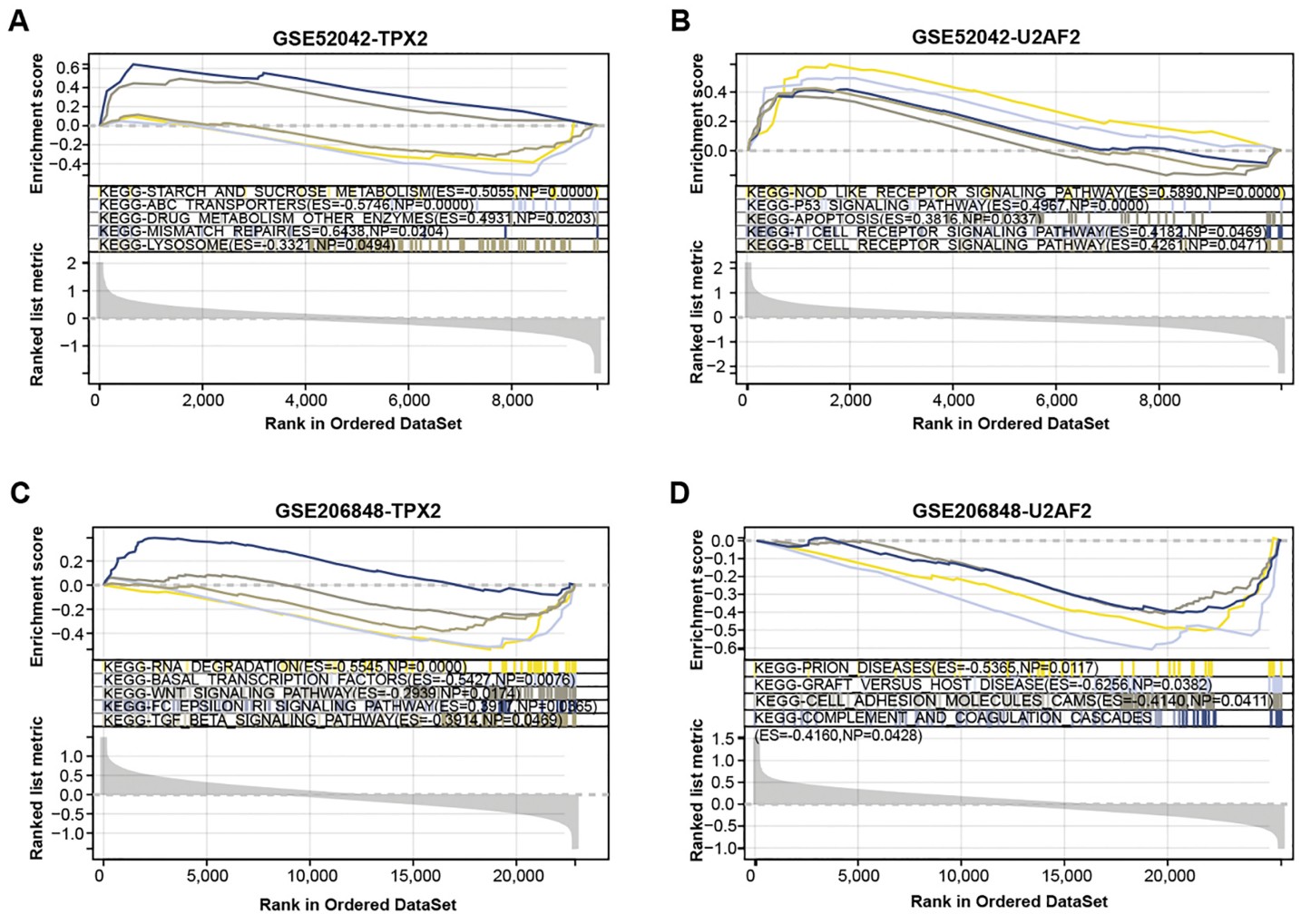

**Figure 3 GSEA-KEGG pathway enrichment analysis of two overlapping genes.** (A–B) KEGG enrichment pathways of TPX2, and U2AF2 in GSE52042 data samples. (C–D) KEGG enrichment pathways of TPX2, and U2AF2 in GSE206848 data samples.

## ROC curve analysis reveals the diagnostic value of two overlapping genes in OA

Based on the "timeROC" package, we conducted ROC curve analysis for the two overlapping genes in GSE52042 and GSE206848 dataset, with results depicted in Figs. 4A–4D. In the GSE52042 dataset, U2AF2 demonstrated the highest AUC value of 1.000, followed closely by TPX2 with an AUC of 0.938. Similarly, in the GSE206848 dataset, U2AF2 exhibited an AUC of 0.889, with TPX2 showing an AUC of 0.873. These results indicate that both genes possess AUC values exceeding 0.7, signifying their robust diagnostic potential for OA. Among them, a research newspaper reported that TPX2 was identified as one of the hub genes in the formation of degenerative meniscal lesions (DML) in OA, indicating that it plays an important role in the underlying molecular mechanism. The current research on U2AF2 in OA is unclear, therefore, we selected TPX2 as the hub gene for the next step of analysis.

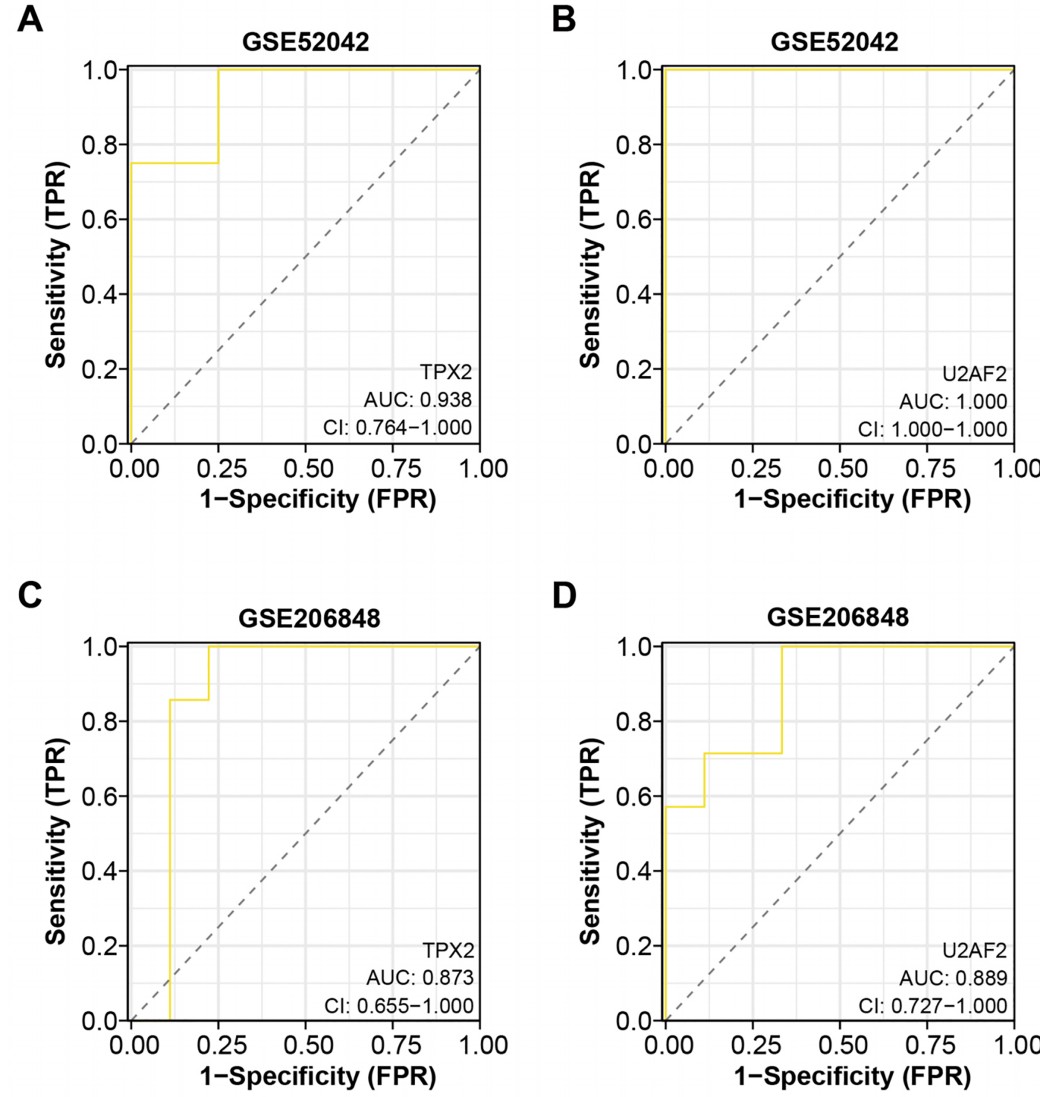

**Figure 4  ROC curve analysis of the two overlapping genes.** (A–B) ROC curve for TPX2, and U2AF2 in GSE52042 with an AUC value of 0.938, and 1.000. Confidence interval (CI) values of 0.764-1.000, 0.764-1.000, and 1.000-1.000. (C–D) ROC curve for TPX2, and U2AF2 in GSE206848 with an AUC value of 0.873, and 0.889. Confidence interval values of 0.579-1.000, 0.655-1.000, and 0.727-1.000.

### Knockdown of TPX2 alleviates LPS-induced damage in C28/I2 cells

TPX2 mRNA and protein expressions were increased in C28/I2 cells treated with LPS at three concentration gradients, as shown in Figs. 5A and 5B. Subsequently, a CCK-8 assay was conducted to detect the proliferative activity of C28/I2 cells with different concentrations of LPS. As indicated in Fig. 5C, cell viability decreased proportionally with higher LPS doses, thus, a concentration of 5 ug/mL LPS was selected for subsequent experiments. After the knockdown of TPX2 in C28/I2 cells treated with 5 ug/mL LPS, qRT-PCR confirmed the efficiency of TPX2 silencing (Fig. 5D). Cell viability assay showed that cell viability was significantly reduced after LPS treatment compared with the negative control, while cells treated with LPS and si-TPX2 showed partial recovery of cell viability

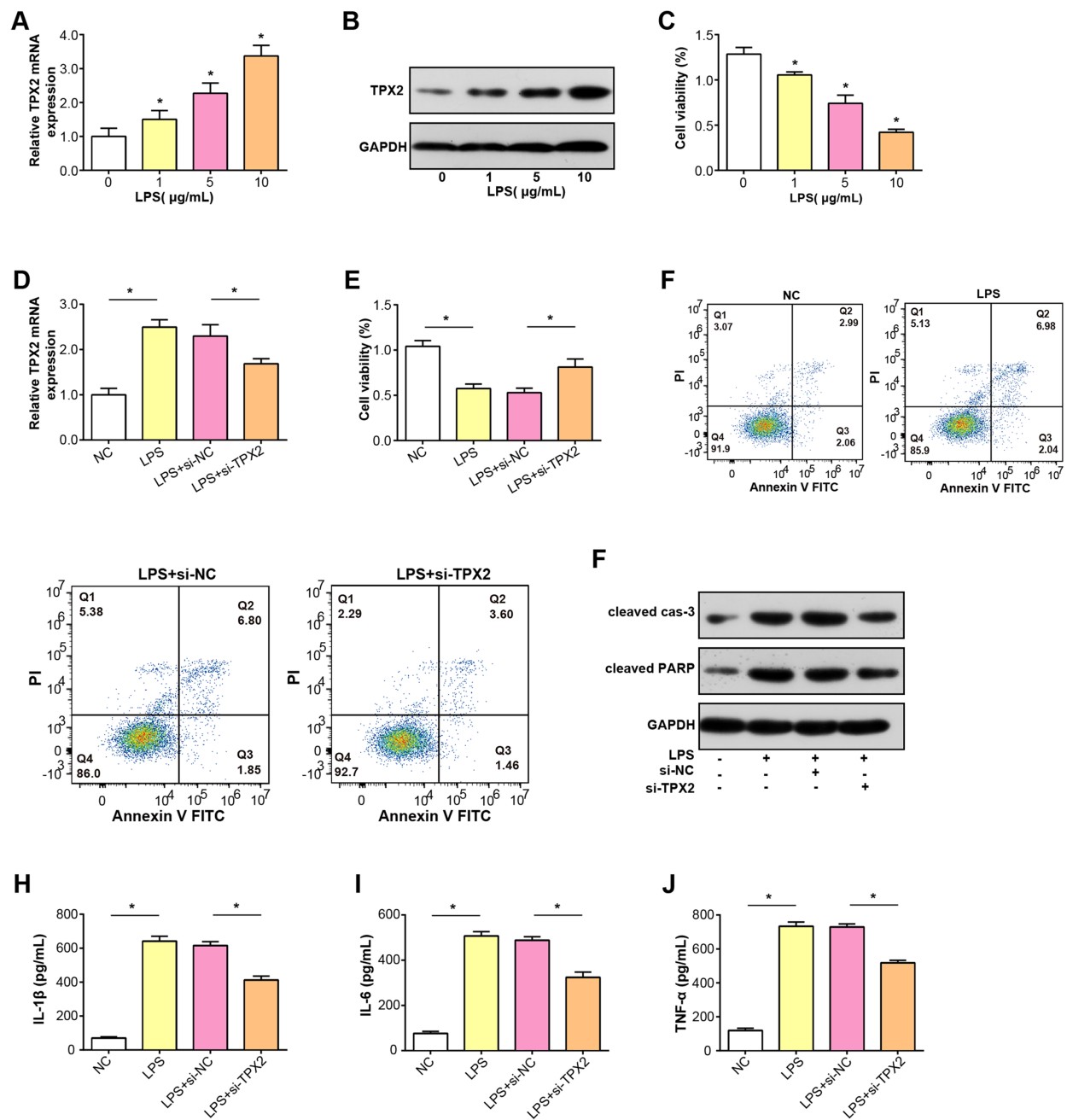

**Figure 5 Effects of LPS treatment and TPX2 knockdown on C28/I2 cells.** (A–B) qRT-PCR and WB were used to detect the expression levels of TPX2 mRNA and protein in C28/I2 cells treated with different concentrations of LPS, respectively. (C) CCK-8 was used to measure the proliferation activity of C28/I2 cells treated with different gradient concentrations of LPS. (D) qRT-PCR detection of TPX2 mRNA expression levels in C28/I2 cells treated with 5ug/mL LPS and TPX2 knockdown. (E) CCK-8 detection of proliferation rate in C28/I2 cells treated with 5ug/mL LPS and TPX2 knockdown. (F) Flow cytometry showed that knockdown of TPX2 can reverse the inhibition of apoptosis in C28/I2 cells induced by LPS treatment. (G) WB detection of expression of apoptosis-related proteins in C28/I2 cells after LPS treatment and knockdown of TPX2. (H–J) ELISA detects the secretion levels of inflammatory cytokines in C28/I2 cells after LPS treatment and TPX2 knockdown. *$P < 0.05$, compared to the control group. Data are expressed as mean ± standard deviation (SD) of three independent experiments technical replicates.

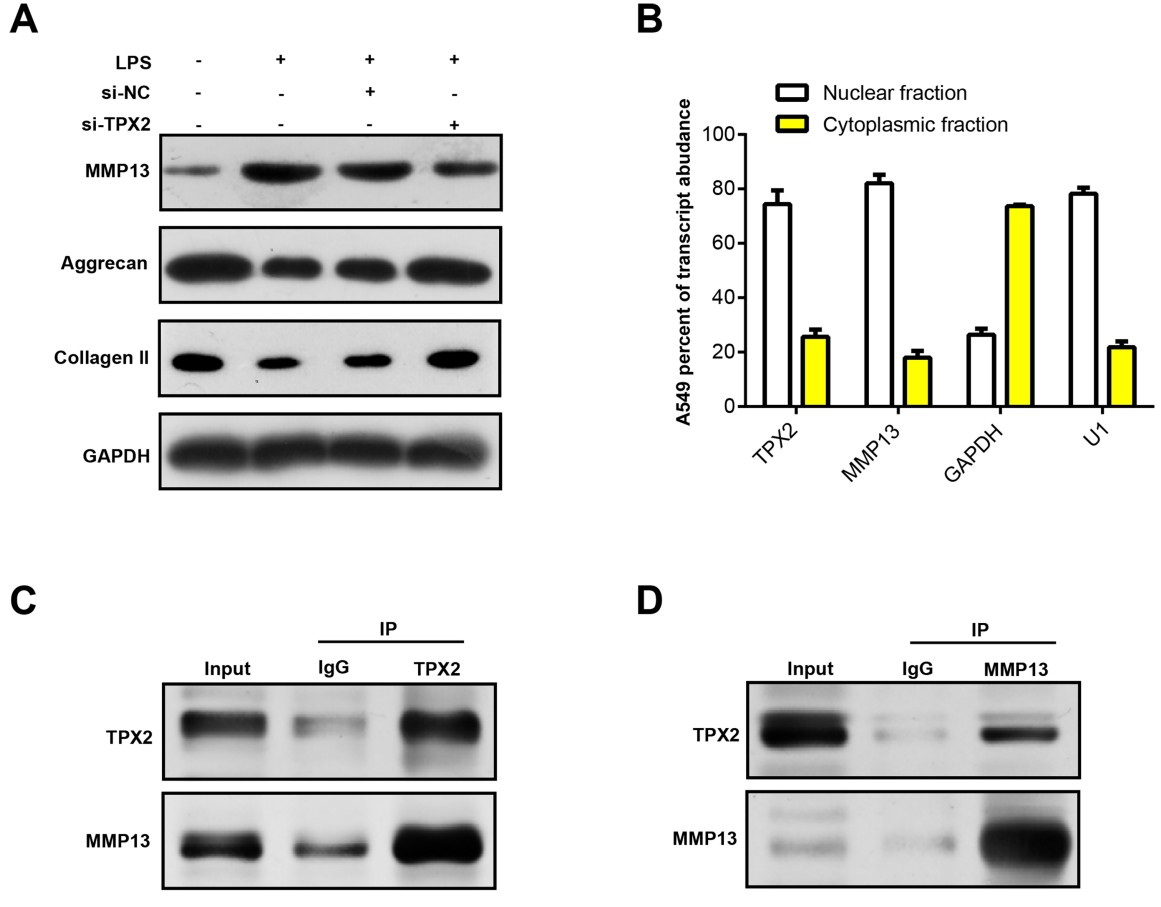

**Figure 6 TPX2 regulates MMP13 and extracellular matrix markers in human normal chondrocytes after LPS treatment.**

(Fig. 5E). The role of TPX2 in LPS-induced cell damage was further investigated by performing apoptosis analysis on LPS-treated C28/I2 cells, and the results showed that TPX2 knockdown inhibited LPS-induced apoptosis (Fig. 5F). WB analysis further supported these findings, with LPS treatment alone increasing levels of cleaved caspase-3 and cleaved PARP (a marker of apoptosis). In contrast, upon knockdown of TPX2, the expression of these apoptotic indicators was reduced compared with cells treated with LPS alone, consistent with flow cytometry data (Fig. 5G). Moreover, ELISA results demonstrated a decrease in the secretion of inflammatory cytokines IL-6, IL-1β, and TNF-α following TPX2 silencing in LPS-induced C28/I2 cells (Figs. 5H–5J). Collectively, these findings suggest that TPX2 knockdown can attenuate LPS-induced damage in C28/I2 cells.

## Protein interaction between TPX2 and MMP13 in OA

MMP13 is an important matrix metalloproteinase involved in biological processes, such as cartilage degradation and joint destruction in OA (*Mehana, Khafaga & El-Blehi, 2019*; *Bouaziz et al., 2016*). Prior research has shown that MMP13 may affect the progression of OA by regulating TPX2 expression (*Alshenibr et al., 2017*). In our study, WB analysis revealed that LPS treatment upregulated MMP13 protein expression in C28/I2 cells and

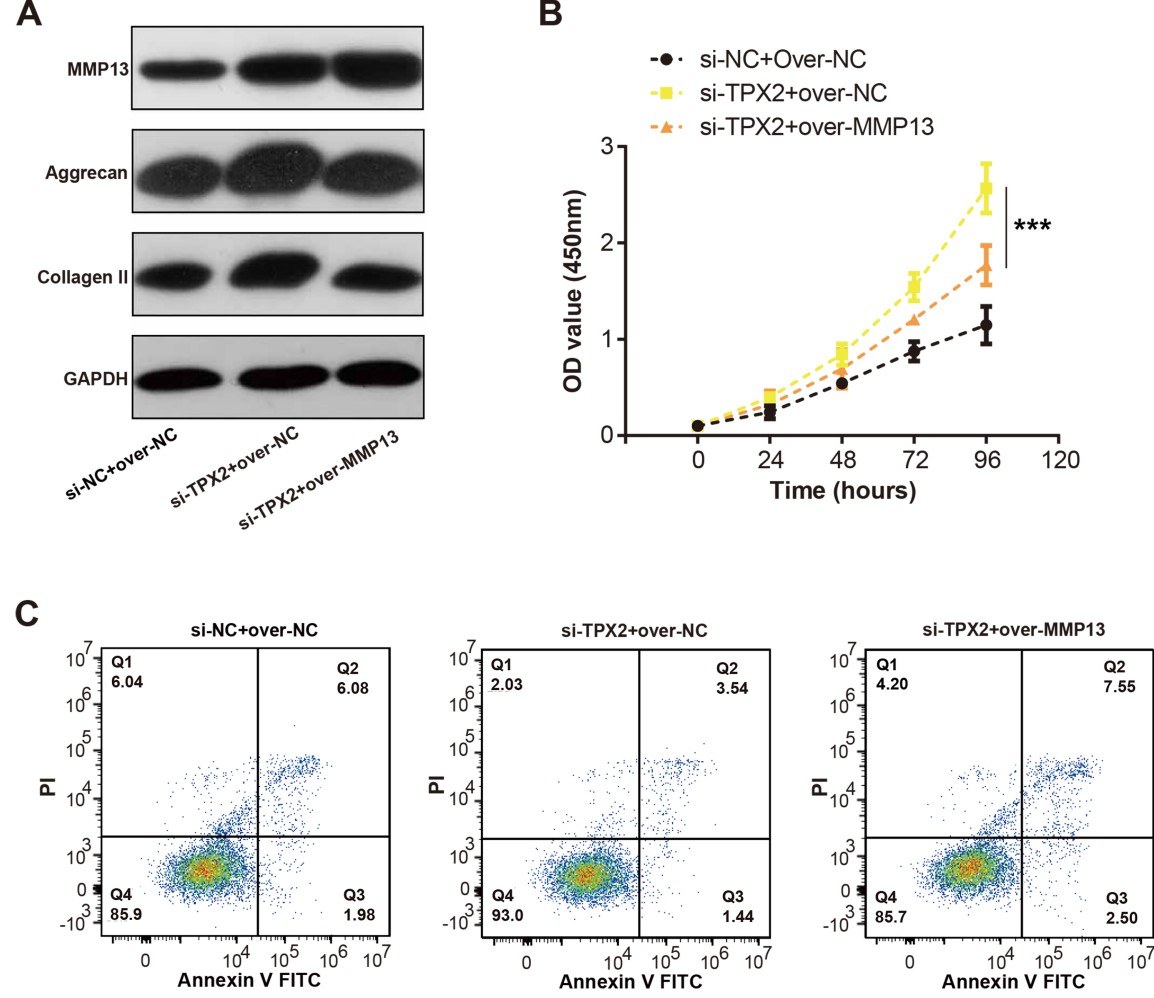

**Figure 7 Effects of TPX2 and MMP-13 on extracellular matrix markers, proliferation, and apoptosis in human normal chondrocytes after LPS treatment.** (A) WB detects the expression of extracellular matrix markers after MMP13 overexpression and TPX2 knockdown. (B) CCK-8 detects the proliferation of C28/I2 cells with TPX2 knockdown and MMP-13 overexpression after LPS treatment. (C) Detection of apoptosis in C28/I2 cells with TPX2 knockdown and MMP-13 overexpression after LPS treatment by flow cytometry. ***$P < 0.001$. Different groups include si-NC + over-NC, si-TPX2 + over-NC and si-TPX2 + over-MMP13. Data are expressed as mean ± standard deviation (SD) of three independent experiments technical replicates.                               

correspondingly diminished extracellular matrix (ECM) markers (Collagen II and Aggrecan). Notably, MMP13 protein levels decreased following TPX2 silencing, whereas Aggrecan and Collagen II levels increased (Fig. 6A). Immunofluorescence staining revealed post-LPS treatment localization of TPX2 and MMP13 predominantly in the nuclei of these cells (Fig. 6B). Subsequently, Co-IP analysis revealed the interaction of TPX2 with MMP13 in C28/I2 cells with LPS treatment (Figs. 6C and 6D). These findings substantiate the protein-protein interaction between TPX2 and MMP13 in OA.

## TPX2 inhibits LPS-induced OA progression through MMP1

We detected the MMP13 and EMC markers levels in C28/I2 cells with si-TPX2 or over-MMP13. As shown in Fig. 7A, TPX2 knockdown resulted in increased levels of MMP13,

Aggrecan and Collagen II protein levels. Conversely, in the si-TPX2+over-MMP13 group, MMP13 protein expressions were up-regulated, while Aggrecan and Collagen II were slightly reduced. Besides, TPX2 knockdown prompted the growth of C28/I2 cells, while MMP13 overexpression attenuated the proliferation activity of C28/I2 cells (Fig. 7B). Flow cytometry assays demonstrated an increase in apoptosis rates in the si-TPX2 and over-MMP13 treated cells (Fig. 7C), indicating that TPX2 plays a crucial role in inhibiting LPS-induced OA progression by targeting MMP13.

## DISCUSSION

OA is a complex joint disease with multifactorial epidemiological factors (*Yoshimura et al., 2017*). Early detection of OA poses challenges, as its diagnosis primarily relies on clinical and radiological manifestations that manifest in the advanced stages of the disease (*Sinusas, 2012*; *Watt, 2018*; *Bielory et al., 2020*). This highlights an urgent need for early diagnostic methods. Microarray technology has been extensively employed to examine gene expression alterations in OA and identify prospective biomarkers (*Aigner et al., 2004*). Further exploration of the molecular mechanisms of OA is critical to deepen our understanding and facilitate the development of new treatment strategies.

In our analysis, 313 DEGs were identified in the GSE52042 dataset, exhibiting significant enrichment in pathways including siRNA binding, VEGF signaling pathway, TNF signaling pathway, malaria, adherens junction, progesterone-mediated oocyte maturation, focal adhesion, platelet activation, and cAMP signaling pathway. These findings suggest potential associations of these genes with OA through the identified pathways. Notably, existing literature links VEGF with the severity of knee osteoarthritis (*Yuan, Sun & Li, 2014*; *Kim et al., 2016*). *Wang et al. (2016)* mentioned that proanthocyanidins could act on OA patients by inhibiting VEFG signaling to provide relief from the onset of the disease. *Yan et al. (2021)* published a description of Wutou Decoction (WTD), an herbal remedy for OA, in which the vascular endothelial growth factor signaling pathway was implicated in the therapeutic process. In a bioinformatics study of synovial inflammation in rheumatoid arthritis, *Xiong et al. (2019)* identified cell adhesion junction as a primary upregulated pathway in DEGs from GSE55235 and GSE55457. *Wang et al. (2015)* also documented that OA-related genes were also significantly enriched in the progesterone-mediated oocyte maturation pathway. Focal adhesion regulates cell adhesion, mechanosensing, and signals that control cell growth and differentiation, acting as cell junctions, adhesions, and support functions (*Yap, Duszyc & Viasnoff, 2018*). Multiple studies have implicated the relation of focal adhesion in the pathogenesis of OA and its association with chondrocyte proliferation and differentiation (*Sang et al., 2021*; *Tsolis et al., 2015*). Besides, other studies have demonstrated the efficacy of platelet-rich plasma in alleviating knee OA pain (*Patel et al., 2013*; *Ayhan, Kesmezacar & Akgun, 2014*).

Through PPI network analysis on DEGs, we screened the top 15 genes with OA-related degree values. By the expression analysis of 15 genes in the OA-related (GSE52042 and GSE206848) datasets, two genes were obtained that were commonly highly expressed in OA samples, namely TPX2, and U2AF2. Then, we used the GSEA method to conduct KEGG pathway enrichment analysis of TPX2, and U2AF2 in the GSE52042 and

GSE206848 datasets, respectively. The significant enrichment pathways included the T cell receptor signaling pathway, RNA degradation, WNT signaling pathway, *etc*. Relevant literature indicates a connection between the T cell receptor signaling pathway and the onset and progression of OA (*Lei et al., 2020*; *Feng & Lian, 2015*). The WNT/β-catenin signaling pathway is known to influence bone, cartilage, and synovial tissue functions in bone and joint pathology (*Zhou et al., 2017*). *Lietman et al. (2018)* also confirmed that blocking of WNT/β-catenin signaling could improve the severity of OA in mice through mouse experiments. *Cong et al. (2021)* showed that T-614 could inhibit joint inflammation by the WNT/β-catenin signaling pathway and inflammatory cytokines. However, the connections between these pathways and OA warrant further exploration and verification.

Subsequently, we constructed ROC curves to assess the clinical diagnostic potential of TPX2, and U2AF2 in OA. Among these genes, U2AF2 exhibited the highest AUC, indicating its significant clinical diagnostic value. *Huan, Jinhe & Rongzong (2019)* reported TPX2 as one of the central genes implicated in degenerative meniscus injury in OA. While U2AF2 has been linked to conditions like frontotemporal dementia and spinocerebellar ataxia 1 (*Richard et al., 2021*; *Paulson et al., 2017*), its relationship with OA remains less defined. The continual exploration and investigation of OA diagnosis and treatment are imperative. Our study leverages bioinformatics for analyzing OA-related biomarkers, thus offering critical insights and underscoring the necessity for continued advancements in OA research.

In our *in vitro* experiments, TPX2 was selected for detailed analysis to elucidate its role in the progression of LPS-induced OA. We observed increased TPX2 mRNA and protein expression under LPS-inducing conditions. Significantly, silencing TPX2 alleviated LPS-induced cellular damage and reduced the secretion of inflammatory markers. The study further uncovered an interaction between TPX2 and MMP13, impacting ECM markers, cell growth, and apoptosis. Knockdown of TPX2 resulted in reduced MMP13 expression and increased levels of ECM components such as aggrecan and collagen II, suggesting a downstream role for MMP13 relative to TPX2. This interaction was substantiated through ELISA assays. Furthermore, TPX2 knockdown enhanced cell proliferation in OA, whereas MMP13 overexpression resulted in a reduced proliferation capacity. Furthermore, increased apoptosis was observed in the TPX2 knockdown combined with MMP13 overexpression group after LPS treatment. Collectively, our findings demonstrate that TPX2 accelerates LPS-induced OA development, with MMP13 potentially acting downstream to regulate ECM homeostasis and cell proliferation.

## CONCLUSION

To sum up, we performed bioinformatic analysis of two OA-related datasets (GSE52042 and GSE206848) and identified two overlapping genes with clinical diagnostic value, namely TPX2 and U2AF2, which are implicated in the pathogenesis of OA. Moreover, *in vitro* experiments verified that TPX2 could act as a hub gene to promote the progression of LPS-induced OA by up-regulating the expression of MMP13 and its interaction with it, which provided some insights for clinical studies. We anticipate that these findings will

provide valuable insights into the treatment and prognosis of OA, thereby offering potential therapeutic avenues for this debilitating condition.

### Funding
The authors received no funding for this work.

### Competing Interests
The authors declare that they have no competing interests.

### Author Contributions
- Jingtao Yu conceived and designed the experiments, prepared figures and/or tables, provided the place, and approved the final draft.
- Weiqi Wang performed the experiments, prepared figures and/or tables, and approved the final draft.
- Zenghui Jiang conceived and designed the experiments, performed the experiments, analyzed the data, authored or reviewed drafts of the article, and approved the final draft.
- Huashun Liu analyzed the data, authored or reviewed drafts of the article, and approved the final draft.

### Data Availability
The raw measurements are available in the Supplemental Files.

### Supplemental Information
Supplemental information for this article can be found online at http://dx.doi.org/10.7717/peerj.17032#supplemental-information.

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
