# Peer review of "TPX2 upregulates MMP13 to promote the progression of lipopolysaccharide-induced osteoarthritis"

_PeerJ, doi:10.7717/peerj.17032_

## Round 0.1 · original submission · Minor Revisions

· Academic Editor

Minor Revisions

The reviewers have highlighted several points for improvement and clarification. In particular, reviewer 2 has identified key issues around the microarray datasets, which are fundamental to the premise of the paper. Can I please ask that you provide detail clarifications in terms of choice of datasets for the differential gene analysis, and the selection of OA samples for analysis, from the overall cohort.

**Language Note:** PeerJ staff have identified that the English language needs to be improved. When you prepare your next revision, please either (i) have a colleague who is proficient in English and familiar with the subject matter review your manuscript, or (ii) contact a professional editing service to review your manuscript. PeerJ can provide language editing services - you can contact us at [email protected] for pricing (be sure to provide your manuscript number and title). – PeerJ Staff

Reviewer 1 ·

Basic reporting

I have a few comments in this section:

In line 15, I would want to use the word “human population” instead of human beings. If you can give the geographical location of the OA would be a plus, for example human population around the world, or human population in “X” continent.

In Line 21, little bit more information on “hub gene” would be helpful.

Line 29, if possible, please try to change this sentence starting with “knocking down….”, looks incomplete. start by something like “By employing bioinformatics techniques, it was seen that the knocking down…”

Line 34, please elaborate the conclusion specially where it states that “some implications”.

Line 45, Please use better sentence where it says, “some previous studies”. It would be nice to cite the names of author or simply say “precious studies”.

In line 85, in my opinion, may be better to use “thereafter” or instead of “afterward”
In line 97, in my opinion, instead of using “after that”, I would start the sentence by saying, “The top 15 genes were then selected from PPI network….”

Everything else looks promising.

Experimental design

Methods described with sufficient detail and information.

Validity of the findings

Data provided are supporting the research and they look promising.

Additional comments

This research is exciting and gives promising results for the research community for further assessments in OA. A few changes suggested in basic reporting should enhance this article.

Reviewer 2 ·

Basic reporting

1. In the background, the authors discuss the next-generation sequencing technology. However, in this manuscript, the two datasets used by the author are microarrays, which is a different technology. The author should add some background information on microarray.
2. The abbreviation "LPS" has not been spelled out for the first occurrence in the abstract and introduction.
3. Please list the R packages and their versions in the manuscript, which is beneficial to reproducibility. I also suggest author provide relevant R scripts.
4. Small numbers ranging from one to ten should generally be spelled out (e.g., lines 73~78)
5. The author does not provide the reference of DAVID (line 90) and GEO2R (line 81).
6. Whether the p-value of the differential gene/enriched GO is original p-value or adjusted p-value (multiple test correction)? The author should provide this information.

Experimental design

1. Both the microarray datasets (GSE52042 and GSE206848) focus on gene expression of OA. Why do authors only use GSE52042 for differential gene expression analysis?
2. Figure 2 E discusses the consistency of the two datasets. However, this figure is limited to 15 genes, providing limited information. I suggest the author compare all differential genes across these two datasets. If one of the datasets suffers from low quality, the author should discard the one and use the high-quality one for other analysis.
3. Before GSE52042 and GSE206848 are used for bioinformatics analysis, the author should check whether there are batch effects in each dataset. If there are batch effects, authors should remove the batch effect to ensure the quality of the analysis.
4. GSE206848 includes 16 samples. The authors only selected seven OA samples and seven normal samples. The author should declare why the remaining two samples should be discarded.

Validity of the findings

The author should check the consistency between the two GEO datasets. If they are not consistent, the author should analyze the reason (e.g., low quality) and do further processing (e.g., remove low-quality dataset). The author should ensure that all results are obtained from consistent datasets to make the story and result convincing.

Reviewer 3 ·

Basic reporting

.

Experimental design

.

Validity of the findings

.

Additional comments

The authors conducted a series of bioinformatics analyses and cell & molecular experiments to explore biomarkers for lipopolysaccharide-induced osteoarthritis, thereby identifying that TPX2 could upregulate MMP13 to promote the progression of lipopolysaccharide-induced osteoarthritis. The current study would be much improved if the authors address the following concerns:


------[Major Concerns about FIGURES, METHODS, RESULTS, and/or CONCLUSIONS]
1. In all FIGURES, it would be clear and more readable to BOTH provide figures with high resolution AND expand on figure legends by explaining the meanings of colors, groups, lines, and abbreviations. These revisions would greatly help readers to understand the results and their implications easily and efficiently. For example,
1.1 In all FIGURES' bar graphs, it would be more informative to display individual data points; in other words, please replace bar graphs by EITHER scatter plots with bars OR scatter plots (a pattern like PMID: 34537192, PMID: 37046252, and PMID: 37452367). Bar graphs have been shown to be misleading, because they cannot reveal variation/dispersion within data; instead, scatter plots with bars could be acceptable and scatter plots would be preferable (as confirmed by PMID: 25901488 and PMID: 28974579).
1.2 In all FIGURES' legends, it would be more rigorous to mention BOTH the sample size (the number of data points OR how many samples/patients were included) AND whether the data points were technical or biological replicates.
1.3 In all FIGURES' legends, it would be more rigorous to mention how the authors reported the data error (variation/dispersion): standard deviation (SD), confidence intervals (CI), or standard error of the mean (SEM, which would be not preferable).

2. In ABSTRACT:
2.1 In Methods, it would be more rigorous and informative to add more details about "GSE52042 and GSE206848" datasets' characteristics: how many samples, what types of samples (controls vs OA?), and what types of tissues.
2.2 In Methods, it would be necessary and more informative to elaborate on what "in vitro experiments" were used. In particular, co-IP and flow cytometry could be highlighted, because they would help readers to understand some text in Results ("TPX2 had a protein interaction", which is usually examined by co-IP).
2.3 In Results ("Through the bioinformatics analysis on GSE52042-DEGs, 3 overlapping genes (NEK2, U2AF2, TPX2) were found with a good clinical diagnostic value, generally highly expressed in the OA samples of GSE52042 and GSE206848, and TPX2 was determined the hub gene"), it would be more rigorous and informative to expand on this sentence by mentioning: 1) the number of up- and down-regulated DEGs in GSE52042, 2) the expression pattern (up- or down-regulated) of the 3 overlapping genes, and 3) specific data showing how "good" is the clinical diagnostic value.
2.4 In Conclusion ("To sum up, TPX2 as an oncogene participates in the progression of LPS-induced OA by up-regulating the expression of MMP13, which provides some implications for clinical research"), it would be more accurate and rigorous to change this sentence into "To sum up, TPX2 could promote the progression of LPS-induced OA by up-regulating the expression of MMP13, which provides some implications for clinical research." Because oncogene seems more widely used in the context of tumor/cancer biology, rather than OA.

3. In INTRODUCTION:
3.1 In Paragraph 2, it would be more informative to re-write this paragraph by replacing it with a section introducing the role of TPX2 & MMP13 in lipopolysaccharide-related disease and/or osteoarthritis. This information seems more valuable than the introduction to sequencing.
3.2 In Paragraph 3, it would be more concise to delete the sentence "The diagnosis and therapy of OA necessitate ongoing exploration and research efforts."

4. In MATERIAL AND METHODS:
4.1 In "Extraction of microarray data information" ("Similarly, another OArelated GSE206848 profile was download, including 16 samples, and in this study we selected 7 normal and 7 OA samples for analysis"), it would be more rigorous to mention why and how the authors "selected" 14 samples out of 16. In other words, please explain why two samples were excluded.
4.2 In "Identification of DEGs related to OA" ("we used fold change (FC)>1.5 as the screening criteria for up-regulation of DEGs, and FC<0.67 for down-regulation of DEGs"), it would be more rigorous to mention & explain why the screening of up- and down-regulated DEGs utilized different FC thresholds.
4.3 In "Construction of Protein-Protein Interaction (PPI)" ("we selected the top 15 genes from the PPI network for the expression analysis in the GSE52042 and GSE206848 datasets"), it would be more rigorous to mention what strategy/algorithm was used to select the "top 15 genes". Cytospace-mediated PPI analysis includes various built-in algorithms for picking out top/hub genes, so please point out the algorithm and threshold that the authors used.
4.4 In "Western blotting (WB)", it would be more rigorous and reproducible to mention antibodies' brands, dilution factors, and species.
4.5 In "Enzyme linked immunosorbent assay (ELISA)", it would be more rigorous and reproducible to mention the ELISA kit's brand.
4.6 In "Immunofluorescence staining", it would be more rigorous and reproducible to mention antibodies' brands, dilution factors, and species.

5. In RESULTS:
5.1 In "Knockdown of TPX2 alleviates LPS-induced damage in C28/I2 cells", before telling "TPX2 mRNA and protein expressions", it would be more rigorous and cohesive (that is, sentences/paragraphs are closely connected) to justify why the authors focused on TPX2 rather than the two other hub genes (NEK2 and/or U2AF2).


------[Minor Concerns about writing]
1. Throughout the manuscript, it seems better to use Grammarly (https://www.grammarly.com/) to check & correct potential grammatical errors or typos. For example,
1.1 In "Extraction of microarray data information" of MATERIAL AND METHODS ("Similarly, another OA related GSE206848 profile was download, including 16 samples, and in this study we selected 7 normal and 7 OA samples for analysis"), it seems better to change this sentence into "Similarly, another OA related GSE206848 profile was downloaded ...".

2. In ABSTRACT:
2.1 In Methods ("Then, the expression analysis on these genes in GSE52042 and GSE206848 was performed for overlapping genes"), it seems better to change this sentence into "Then, the expression of these genes in GSE52042 and GSE206848 was analyzed to identify overlapping genes". After this revision, the sentence would be more concise and clearer (easier to understand).
2.2 In Methods ("Next, hub gene was determined after Gene Set Enrichment Analysis (GSEA) and clinical diagnostic value analysis on the overlapping genes"), it seems better to change this sentence into "Next, the overlapping genes were scrutinized by Gene Set Enrichment Analysis (GSEA) and clinical diagnostic value analysis to single out the hub gene." After this revision, the sentence would be more cohesive (that is, sentences are closely connected).
2.3 In Methods ("Finally, in vitro experiments verified the function and mechanism of hub gene on LPS-induced OA progression"), it seems better to change this sentence into "Finally, whether and how the hub gene impacts LPS-induced OA progression was explored by in vitro experiments." After this revision, the sentence would be more cohesive.

---

## Round 0.2 · Major Revisions

· Academic Editor

Major Revisions

Following a second round of review, the referees have identified several areas for improvement. In particular, reviewer 3 has highlighted numerous points of improvement for the manuscript - please address these accordingly.

Reviewer 1 ·

Basic reporting

The edits in the revision is satisfactory and adds the value to your article with clear understanding. No further comments.

Experimental design

Research is designed with sufficient details. No further comments.

Validity of the findings

Study is well concluded supported with sufficient data. No further comments.

Additional comments

No comment.

Reviewer 2 ·

Basic reporting

The issues have been addressed.

Experimental design

The issues have been addressed.

Validity of the findings

The author does not check the batch effects of the two datasets. I suggest the authors check it simply using a heatmap.

Reviewer 3 ·

Basic reporting

Please see the Additional Comments.

Experimental design

Please see the Additional Comments.

Validity of the findings

Please see the Additional Comments.

Additional comments

The authors conducted a series of bioinformatics analyses and cell & molecular experiments to explore biomarkers for lipopolysaccharide-induced osteoarthritis, thereby identifying that TPX2 could upregulate MMP13 to promote the progression of lipopolysaccharide-induced osteoarthritis. The current study would be much improved if the authors address the following concerns:


------[Major Concerns about FIGURES, METHODS, RESULTS, and/or CONCLUSIONS]
1. In all FIGURES, it would be clear and more readable to BOTH provide figures with high resolution AND expand on figure legends by explaining the meanings of colors, groups, lines, and abbreviations. These revisions would greatly help readers to understand the results and their implications easily and efficiently. For example,
1.1 In all FIGURES' bar graphs, it would be more informative to display individual data points; in other words, please replace bar graphs by EITHER scatter plots with bars OR scatter plots (a pattern like PMID: 34537192, PMID: 37046252, and PMID: 37452367). Bar graphs have been shown to be misleading, because they cannot reveal variation/dispersion within data; instead, scatter plots with bars could be acceptable and scatter plots would be preferable (as confirmed by PMID: 25901488 and PMID: 28974579).
1.2 In all FIGURES' legends, it would be more rigorous to mention BOTH the sample size (the number of data points OR how many samples/patients were included) AND whether the data points were technical or biological replicates.
1.3 In all FIGURES' legends, it would be more rigorous to mention how the authors reported the data error (variation/dispersion): standard deviation (SD), confidence intervals (CI), or standard error of the mean (SEM, which would be not preferable).

2. In ABSTRACT:
2.1 In Methods, it would be more rigorous and informative to add more details about "GSE52042 and GSE206848" datasets' characteristics: how many samples, what types of samples (controls vs OA?), and what types of tissues.
2.2 In Methods, it would be necessary and more informative to elaborate on what "in vitro experiments" were used. In particular, co-IP and flow cytometry could be highlighted, because they would help readers to understand some text in Results ("TPX2 had a protein interaction", which is usually examined by co-IP).
2.3 In Results ("Through the bioinformatics analysis on GSE52042-DEGs, 3 overlapping genes (NEK2, U2AF2, TPX2) were found with a good clinical diagnostic value, generally highly expressed in the OA samples of GSE52042 and GSE206848, and TPX2 was determined the hub gene"), it would be more rigorous and informative to expand on this sentence by mentioning: 1) the number of up- and down-regulated DEGs in GSE52042, 2) the expression pattern (up- or down-regulated) of the 3 overlapping genes, and 3) specific data showing how "good" is the clinical diagnostic value.
2.4 In Conclusion ("To sum up, TPX2 as an oncogene participates in the progression of LPS-induced OA by up-regulating the expression of MMP13, which provides some implications for clinical research"), it would be more accurate and rigorous to change this sentence into "To sum up, TPX2 could promote the progression of LPS-induced OA by up-regulating the expression of MMP13, which provides some implications for clinical research." Because oncogene seems more widely used in the context of tumor/cancer biology, rather than OA.

3. In INTRODUCTION:
3.1 In Paragraph 2, it would be more informative to re-write this paragraph by replacing it with a section introducing the role of TPX2 & MMP13 in lipopolysaccharide-related disease and/or osteoarthritis. This information seems more valuable than the introduction to sequencing.
3.2 In Paragraph 3, it would be more concise to delete the sentence "The diagnosis and therapy of OA necessitate ongoing exploration and research efforts."

4. In MATERIAL AND METHODS:
4.1 In "Extraction of microarray data information" ("Similarly, another OArelated GSE206848 profile was download, including 16 samples, and in this study we selected 7 normal and 7 OA samples for analysis"), it would be more rigorous to mention why and how the authors "selected" 14 samples out of 16. In other words, please explain why two samples were excluded.
4.2 In "Identification of DEGs related to OA" ("we used fold change (FC)>1.5 as the screening criteria for up-regulation of DEGs, and FC<0.67 for down-regulation of DEGs"), it would be more rigorous to mention & explain why the screening of up- and down-regulated DEGs utilized different FC thresholds.
4.3 In "Construction of Protein-Protein Interaction (PPI)" ("we selected the top 15 genes from the PPI network for the expression analysis in the GSE52042 and GSE206848 datasets"), it would be more rigorous to mention what strategy/algorithm was used to select the "top 15 genes". Cytospace-mediated PPI analysis includes various built-in algorithms for picking out top/hub genes, so please point out the algorithm and threshold that the authors used.
4.4 In "Western blotting (WB)", it would be more rigorous and reproducible to mention antibodies' brands, dilution factors, and species.
4.5 In "Enzyme linked immunosorbent assay (ELISA)", it would be more rigorous and reproducible to mention the ELISA kit's brand.
4.6 In "Immunofluorescence staining", it would be more rigorous and reproducible to mention antibodies' brands, dilution factors, and species.


5. In RESULTS:
5.1 In "Knockdown of TPX2 alleviates LPS-induced damage in C28/I2 cells", before telling "TPX2 mRNA and protein expressions", it would be more rigorous and cohesive (that is, sentences/paragraphs are closely connected) to justify why the authors focused on TPX2 rather than the two other hub genes (NEK2 and/or U2AF2).

------[Minor Concerns about writing]
1. Throughout the manuscript, it seems better to use Grammarly (https://www.grammarly.com/) to check & correct potential grammatical errors or typos. For example,
1.1 In "Extraction of microarray data information" of MATERIAL AND METHODS ("Similarly, another OA related GSE206848 profile was download, including 16 samples, and in this study we selected 7 normal and 7 OA samples for analysis"), it seems better to change this sentence into "Similarly, another OA related GSE206848 profile was downloaded ...".

2. In ABSTRACT:
2.1 In Methods ("Then, the expression analysis on these genes in GSE52042 and GSE206848 was performed for overlapping genes"), it seems better to change this sentence into "Then, the expression of these genes in GSE52042 and GSE206848 was analyzed to identify overlapping genes". After this revision, the sentence would be more concise and clearer (easier to understand).
2.2 In Methods ("Next, hub gene was determined after Gene Set Enrichment Analysis (GSEA) and clinical diagnostic value analysis on the overlapping genes"), it seems better to change this sentence into "Next, the overlapping genes were scrutinized by Gene Set Enrichment Analysis (GSEA) and clinical diagnostic value analysis to single out the hub gene." After this revision, the sentence would be more cohesive (that is, sentences are closely connected).
2.3 In Methods ("Finally, in vitro experiments verified the function and mechanism of hub gene on LPS-induced OA progression"), it seems better to change this sentence into "Finally, whether and how the hub gene impacts LPS-induced OA progression was explored by in vitro experiments." After this revision, the sentence would be more cohesive.

---

## Round 0.3 · accepted · Accept

· Academic Editor

Accept

Having reviewed the article, the last round of comments have been adequately addressed by the authors, and I am satisfied it is ready for publication.